# Towards Training One-Step Diffusion Models Without Distillation

## Abstract

Recent advances in training one-step diffusion models typically follow a two-stage pipeline: first training a teacher diffusion model and then distilling it into a one-step student model. This process often depends on both the teacher's score function for supervision and its weights for initializing the student model. In this paper, we explore whether one-step diffusion models can be trained directly without this distillation procedure. We introduce a family of new training methods that entirely forgo teacher score supervision, yet outperforms most teacher-guided distillation approaches. This suggests that score supervision is not essential for effective training of one-step diffusion models. However, we find that initializing the student model with the teacher's weights remains critical. Surprisingly, the key advantage of teacher initialization is not due to better latent-to-output mappings, but rather the rich set of feature representations across different noise levels that the teacher diffusion model provides. These insights take us one step closer towards training one-step diffusion models without distillation and provide a better understanding of the roles of teacher supervision and initialization in the distillation process.

## 1 Introduction

Diffusion models (Sohl-Dickstein et al., 2015; Ho et al., 2020; Song & Ermon, 2019) have achieved remarkable success in modeling complex real-world data across a wide range of domains, including image synthesis (Rombach et al., 2022; Li et al., 2022), 3D generation (Poole et al., 2022a), video synthesis (Ho et al., 2022), equivariant modeling (Hoogeboom et al., 2022), and audio generation (Liu et al., 2023). Typically, diffusion models consist of two processes: a forward noising process, which gradually perturbs data into a known noise prior (typically Gaussian noise), and a reverse denoising process, which learns to invert the forward corruption process to generate realistic data samples from noise. Formally, the forward process is defined over $T$ time steps as a Markov chain of Gaussian transitions, while the reverse process is parameterized using neural networks that predict the denoising distribution given the noisy samples.

In classic diffusion models with Gaussian denoising distributions, generating high-quality data samples typically requires hundreds or thousands of sampling steps, resulting in significant sampling inefficiency due to the need for $T \gg 1$ NFEs (number of function evaluations) (Ho et al., 2020; Nichol & Dhariwal, 2021). To address this weakness, various acceleration methods have been proposed to reduce NFEs during sampling. One class of such approaches leverages advanced numerical solvers for differential equations, enabling continuous-time approximations of the diffusion process (Song et al., 2020; Liu et al., 2022; Lu et al., 2022). Another line of work improves the flexibility of the posterior distribution in the denoising process, either by estimating a more accurate covariance for the Gaussian distribution (Nichol & Dhariwal, 2021; Bao et al., 2022b;a; Ou et al., 2025) or by adopting flexible non-Gaussian denoising distributions (Bortoli et al., 2025; Xiao et al., 2021; Yu et al., 2024). While these techniques can dramatically reduce NFEs from $\sim 10^3$ to around 10–20, they still fall short of achieving high-quality generation within 5 steps.

Recently, distillation-based methods have emerged as a powerful direction for training diffusion models, enabling high-quality *one-step* generation (Zhou et al., 2024). These methods fall into two categories:

- **Trajectory-based distillation methods** (Salimans & Ho, 2022; Berthelot et al., 2023; Song et al., 2023; Heek et al., 2024; Kim et al., 2024; Li & He, 2024) aim to approximate the full sampling trajectory by training a student model to amortize multiple intermediate steps. These methods are motivated by accelerated solvers and typically perform joint training of the full diffusion model and the distillation process.

- **Score-based distillation approaches** (Luo et al., 2024; Salimans et al., 2024; Xie et al., 2024; Zhou et al., 2024) distill the full denoising process of the pre-trained teacher diffusion model into a one-step latent variable model. This distillation process typically involves minimizing the divergence between the student and teacher models based on their respective score estimations (Poole et al., 2022a; Wang et al., 2024).

In this paper, we focus on score-based distillation methods and investigate whether a one-step diffusion model can be effectively trained without relying on a pre-trained teacher diffusion model. In existing distillation approaches, the teacher model is typically used in two key places: (1) the teacher's score function is used to estimate the gradient for training the student model, and (2) the teacher's weights are used to initialize the student model. The goal of this paper is to investigate whether it is possible to train a one-step diffusion model without using either the teacher's score function or its weight initialization.

Our main contributions are summarized as follows:

1. We propose a novel distillation method that eliminates the need for both teacher and student score estimation during training. Despite this simplification, our method outperforms most one-step generation approaches and achieves competitive performance to the state-of-the-art methods on image generation tasks without the supervision of the teacher score function.

2. We further analyze the importance of initializing the student model with teacher model's weights from both the weight-space and function-space perspectives, providing deeper insights into the role of teacher weight initialization. This analysis lays the groundwork for future efforts toward training one-step diffusion models entirely without reliance on a teacher model.

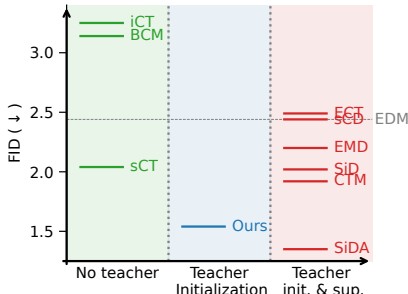

Figure 1: FID comparison of one-step generation on ImageNet $64\times64$. Our method achieves competitive performance to the state-of-the-art method without the supervision of the teacher model (EDM)'s score function.

Before introducing our proposed method, we will first establish the background on diffusion models and score-based distillation methods in the next section.

## 2 BACKGROUND

### 2.1 DENOISING DIFFUSION MODELS

Let $\{x^{(1)}, \ldots, x^{(N)}\}$ denote data samples from the true data distribution $p_d(x_0)$. Diffusion models (Sohl-Dickstein et al., 2015; Ho et al., 2020; Song & Ermon, 2019) define a generative process that transforms samples from a simple Gaussian prior $p(x_T)$ into complex data distributions $p(x_0)$ through a learned denoising process. The model consists of two main components: a forward noising process and a reverse denoising process.

The forward process defines a Markov chain that progressively adds Gaussian noise to the data:

$$q(x_{0:T}) = p_d(x_0) \prod_{t=1}^{T} q(x_t|x_{t-1}), \tag{1}$$

with transition kernels defined as

$$q(x_t|x_{t-1}) = \mathcal{N}(x_t|\sqrt{1-\beta_t}\,x_{t-1}, \beta_t I), \tag{2}$$

where $\beta_t \in (0, 1)$ is a pre-specified variance schedule. This process gradually perturbs the data until it resembles an isotropic Gaussian distribution. The skip distribution at time $t$ can be written as:

$$q(x_t|x_0) = \mathcal{N}(x_t|\sqrt{\bar{\alpha}_t}x_0, (1 - \bar{\alpha}_t)I), \tag{3}$$

where $\bar{\alpha}_t = \prod_{s=1}^{t}(1 - \beta_s)$. As $T \to \infty$, the final state $x_T$ approximates a standard normal distribution, i.e., $q(x_T) \to \mathcal{N}(0, I)$.

The generative process aims to reverse this trajectory. Starting from noise $x_T \sim p(x_T) = \mathcal{N}(0, I)$, the model learns a reverse process to sequentially denoise and reconstruct data samples. Since the true reverse conditional $q(x_{t-1}|x_t)$ is intractable, a common method is to approximate it with a variational Gaussian distribution:

$$p_\theta(x_{t-1}|x_t) = \mathcal{N}(x_{t-1}|\mu_{t-1}(x_t; \theta), \Sigma_{t-1}(x_t; \theta)), \tag{4}$$

where the mean function is learned from data and the covariance function can be either learned (Nichol & Dhariwal, 2021; Ou et al., 2025; Bao et al., 2022a) or chosen to be a fixed value (Bao et al., 2022b; Ho et al., 2020).

From a score-based perspective, learning the mean function $\mu_{t-1}(x_t; \theta)$ is equivalent to learning the score function $\nabla_{x_t} \log q(x_t)$, which represents the gradient of the log-density of the noised data $x_t$. This score can be approximated using denoising score matching (DSM) (Vincent, 2011; Song & Ermon, 2019), and transformed into the reverse mean through Tweedie's Lemma (Efron, 2011; Robbins, 1992):

$$\mu_{t-1}(x_t; \theta) = \frac{1}{\sqrt{1 - \beta_t}} \left( x_t + \beta_t \nabla_{x_t} \log p_\theta(x_t) \right), \tag{5}$$

where $\nabla_{x_t} \log p_\theta(x_t) \approx \nabla_{x_t} \log q(x_t)$. This establishes a connection between the denoising distributional perspective and the score estimation perspective of diffusion models. In the following section, we introduce score-based distillation methods through the lens of divergence minimization.

## 2.2 SCORE-BASED DISTILLATION METHODS

Score-based distillation methods aim to distill a teacher diffusion model $p_\theta$ (pre-trained on the true data distribution $p_d(x_0)$) into a one-step implicit generative model (Goodfellow et al., 2014; Huszár, 2017; Zhang et al., 2020):

$$q_\theta(x_0) = \int \delta(x_0 - g_\theta(z))p(z)dz, \tag{6}$$

where $\delta(\cdot)$ is the Dirac delta function, $p(z)$ is a standard Gaussian prior for the latent variable $z$, and $g_\theta : \mathcal{Z} \to \mathcal{X}$ is a deterministic neural network that generates data $x$ from the latent variable $z$ in one step. We emphasize that when the function $g_\theta(\cdot)$ is not bijective, the model distribution $q_\theta$ is not absolutely continuous with respect to the Lebesgue measure. As a result, the corresponding density function may not be well-defined, and consequently, the KL divergence between $q_\theta(x_0)$ and the data distribution $p_d(x_0)$ may also be ill-defined (Arjovsky et al., 2017; Zhang et al., 2020).

Inspired by diffusion models, one can use a set of (scaled) Gaussian convolution kernels $\mathcal{K} = \{k_1, \cdots, k_T\}$ defined by $k_t(x_t|x_0) = \mathcal{N}(x_t|\alpha_t x_0, \sigma_t^2 I)$ to define the Diffusive KL divergence (DiKL) between the model density $q_\theta(x_0)$ and target distribution $p_d(x_0)$:

$$\mathrm{DiKL}_{\mathcal{K}}(q_\theta(x_0)||p_d(x_0)) \equiv \sum_{t=1}^{T} w(t)\mathrm{KL}(q_\theta(x_t)||p_d(x_t)), \tag{7}$$

where $w(t)$ is a positive scalar weighting function that sums to one, and $q_\theta(x_t)$ and $p_d(x_t)$ are noisy model density and noisy target density, respectively, defined by

$$q_\theta(x_t) = \int q_\theta(x_0)k_t(x_t|x_0)dx_0 \quad \text{and} \quad p_d(x_t) = \int p_d(x_0)k_t(x_t|x_0)dx_0. \tag{8}$$

In this case, the model distribution $q_\theta(x_t)$ is always absolutely continuous, and thus the KL divergence between them is always well-defined. For a single Gaussian kernel, the divergence was previously known as *Spread KL divergence* (Zhang et al., 2020; 2019). It is straightforward to show

---

**Algorithm 1** Score-Based Distillation of One-Step Diffusion Models

---

**Require:** Data samples $\{x^{(1)}, \ldots, x^{(N)}\} \sim p_d(x_0)$

———————————— Stage 1: Train a multi-step teacher diffusion model ————————————

1: Train the teacher model's score network $s^{p_d}_{\psi_1}(x_t, t)$ using DSM (Eq. 10) until convergence

———————————— Stage 2: Train a one-step student generative model ————————————

2: Initialize the one-step generator with the teacher's score network $g_{\theta_{\text{init}}}(\cdot) \equiv s^{p_d}_{\psi_1}(\cdot, t = t_{\text{init}})$
3: **for** each training iteration **do**
4:     Estimate the student model's score by training a score network $s^{q_\theta}_{\psi_2}(x_t, t)$ using DSM
5:     Update the one-step generator $g_\theta$ with $s^{q_\theta}_{\psi_2}(x_t, t)$ and $s^{p_d}_{\psi_1}(x_t, t)$ using Eq. 12
6: **end for**

---

that it is a valid divergence, i.e., $\text{DiKL}_{\mathcal{K}}(q_\theta || p_d) = 0 \Leftrightarrow q_\theta = p_d$; see Zhang et al. (2020) for a proof. In addition to the diffusion distillation (Luo et al., 2024; Xie et al., 2024), this divergence has successfully been used in 3D generative models (Poole et al., 2022b; Wang et al., 2024) and training neural samplers (He et al., 2024).

Without loss of generality, we consider a single Gaussian convolution kernel $k_t$. The gradient of DiKL with respect to the model parameters $\theta$ can be obtained analytically (He et al., 2024):

$$\nabla_\theta \text{KL}(q_\theta(x_t) || p_d(x_t)) = \int q_\theta(x_t) \left( \nabla_{x_t} \log q_\theta(x_t) - \nabla_{x_t} \log p_d(x_t) \right) \frac{\partial x_t}{\partial \theta} dx_t. \tag{9}$$

However, neither the noisy model score $\nabla_{x_t} \log q_\theta(x_t)$ nor the noisy target score $\nabla_{x_t} \log p_d(x_t)$ are directly accessible. In the distillation setting, the noisy target score is provided by a pre-trained diffusion model, which has been trained using *denoising score matching* (DSM) (Vincent, 2011). Specifically, the score network $s^{p_d}_{\psi_1}(x_t, t) \approx \nabla_{x_t} \log p_d(x_t)$ provides an estimate of the noisy data score based on access to samples from $p_d(x_t)$ and the tractable score $\nabla_{x_t} \log k_t(x_t \mid x_0)$, which is learned by

$$\min_{\psi_1} \mathcal{L}_{\text{DSM}}(\psi_1) = \frac{1}{2} \iint \| s^{p_d}_{\psi_1}(x_t, t) - \nabla_{x_t} \log k_t(x_t|x_0) \|^2_2 p_d(x_0) p(x_t|x_0) dx_t dx_0. \tag{10}$$

Regarding the noisy model score $\nabla_{x_t} \log q_\theta(x_t)$, we note that since we can efficiently sample from the one-step student model $q_\theta(x_t)$, we can approximate its score function using another score network $s^{q_\theta}_{\psi_2}(x_t, t) \approx \nabla_{x_t} \log q_\theta(x_t)$, trained with the DSM loss:

$$\min_{\psi_2} \mathcal{L}_{\text{DSM}}(\psi_2) = \frac{1}{2} \iint \| s^{q_\theta}_{\psi_2}(x_t, t) - \nabla_{x_t} \log k_t(x_t|x_0) \|^2_2 q_\theta(x_0) p(x_t|x_0) dx_t dx_0. \tag{11}$$

Thus, the gradient of DiKL with respect to the parameters $\theta$ of the student model can be estimated as follows, a method known as Variational Score Distillation (VSD) (Poole et al., 2022a; Wang et al., 2024; Luo et al., 2024):

$$\nabla_\theta \text{DiKL}(q_\theta(x_0) || p_d(x_0)) \approx \sum_{t=1}^{T} w(t) \int q_\theta(x_t) \left( s^{q_\theta}_{\psi_2}(x_t, t) - s^{p_d}_{\psi_1}(x_t, t) \right) \frac{\partial x_t}{\partial \theta} dx_t. \tag{12}$$

Unlike the teacher score function which remains fixed after pre-training, the noisy model score $\nabla_{x_t} \log q_\theta(x_t)$ dynamically changes as we update the student model's parameters $\theta$ during training. Therefore, the score network $s^{q_\theta}_{\psi_2}(x_t, t) \approx \nabla_{x_t} \log q_\theta(x_t)$ needs to be updated every time we update the student model, which results in an interleaved training procedure as detailed in Algorithm 1.

We observe that training with DiKL typically requires estimating the student model score and using the teacher score for supervision. In the next section, we propose a method that enables training one-step diffusion models without relying on student score estimation or teacher score supervision.

## 3 Training One-Step Diffusion Without Score Distillation

We first explore whether a one-step diffusion model can be trained without teacher supervision (i.e., without relying on pre-trained teacher score, denoiser, ODE solver, etc.). Starting from Algorithm

---

**Algorithm 2** Score-free Training of One-Step Diffusion Models

---

**Require:** Data samples $\{x^{(1)}, \dots, x^{(N)}\} \sim p_d(x_0)$

———————————— Stage 1: Train a multi-step teacher diffusion model ————————————

1: Train the teacher model's score network $s_{\psi_1}^{p_d}(x_t, t)$ using DSM (Eq. 10) until convergence

———————————— Stage 2: Train a one-step student generative model ————————————

2: Initialize the one-step generator with the teacher's score network $g_{\theta_{\text{init}}}(\cdot) \equiv s_{\psi_1}^{p_d}(\cdot, t = t_{\text{init}})$
3: **for** each training iteration **do**
4:     Estimate the density ratio $q_\theta(x_t)/p_d(x_t)$ by training a neural network classifier $c_\eta(x_t, t)$
5:     Update the one-step generator $g_\theta$ with $c_\eta(x_t, t)$ using Eq. 17 or Eq. 18 or Eq. 19
6: **end for**

---

1, we note that the DiKL gradient estimator relies on the score difference, $s_{\psi_1}^{q_\theta}(x_t, t) - s_{\psi_2}^{p_d}(x_t, t)$. To eliminate the dependency on the teacher's score function $s_{\psi_2}^{p_d}(x_t, t)$, we first observe that the score difference can be written as the gradient of a log-density-ratio:

$$\nabla_{x_t} \log q_\theta(x_t) - \nabla_{x_t} \log p_d(x_t) = \nabla_{x_t} \log(q_\theta(x_t)/p_d(x_t)). \tag{13}$$

Therefore, rather than estimating the two scores separately, we directly estimate the density ratio between the student and teacher at all noise levels using class-ratio estimation (Sugiyama et al., 2012; Qin, 1998; Gutmann & Hyvärinen, 2010; Zhang et al., 2022).

### 3.1 CLASS-RATIO ESTIMATION

We first denote distributions $q_\theta(x_t)$ and $p_d(x_t)$ as two conditional distributions $m(x_t|y = 0)$ and $m(x_t|y = 1)$, respectively, where $y = 0$ indicates samples from the student model $q_\theta(x_t)$ and $y = 1$ indicates data samples from $p_d(x_t)$. With Bayes' rule, we can transform the density ratio estimation problem into a binary classification problem:

$$\frac{q_\theta(x_t)}{p_d(x_t)} \equiv \frac{m(x_t|y = 0)}{m(x_t|y = 1)} = \frac{p(y = 0|x_t)m(x_t)}{p(y = 0)} \Big/ \frac{p(y = 1|x_t)m(x_t)}{p(y = 1)} = \frac{p(y = 0|x_t)}{p(y = 1|x_t)}, \tag{14}$$

where the mixture distribution is defined as

$$m(x) \equiv m(x_t|y = 1)p(y = 1) + m(x_t|y = 0)p(y = 0), \tag{15}$$

and the Bernoulli prior distribution $p(y)$ is simply set as a uniform prior $p(y = 1) = p(y = 0) = 0.5$. In practice, we sample a batch of data from $p_d(x_t)$ and assign them the label $y = 0$, and sample an equal number of samples from $q_\theta(x_t)$, assigning them the label $y = 1$. We then train a neural network classifier $c_\eta(x_t, t)$, conditioned on the diffusion time $t$, to estimate the probability that a given input $x_t$ belongs to class $y = 1$. The optimal classifier approximates the posterior probability $c^*(x_t, t) = p(y = 1 \mid x_t, t)$. In this case, the log-density ratio can be estimated as

$$\nabla_{x_t} \log \frac{q_\theta(x_t)}{p_d(x_t)} \approx \nabla_{x_t} \log \frac{1 - c_\eta(x_t, t)}{c_\eta(x_t, t)} = \nabla_{x_t} \text{logit}(1 - c_\eta(x_t, t)). \tag{16}$$

Estimating the density ratio in the noisy space has the advantage of increasing the overlap between the supports of the two distributions, thereby stabilizing the training process.

Importantly, our method does not require any forms of teacher score supervision, as it avoids the need of using the teacher score $s_{\psi_1}^{q_\theta}(x_t, t)$ to approximate the noisy data score $\nabla_{x_t} \log p_d(x_t)$. Furthermore, compared to VSD, our approach employs a single class-ratio estimator, which is more memory-efficient and consistent than the two independently trained score networks $s_{\psi_1}^{q_\theta}$ and $s_{\psi_2}^{p_d}$ used in the original VSD loss (Equation 12).

### 3.2 CLASS-RATIO GRADIENT ESTIMATORS FOR TRAINING ONE-STEP DIFFUSION MODELS

We can then obtain a new gradient estimator for DiKL by applying our class-ratio estimator to Equation 9:

$$\nabla_\theta \text{DiKL}(q_\theta(x_0)||p_d(x_0)) \approx \sum_{t=1}^{T} w(t) \int q_\theta(x_t) \nabla_{x_t} \text{logit}(1 - c_\eta(x_t, t)) \frac{\partial x_t}{\partial \theta} dx_t. \tag{17}$$

In addition to the DiKL, we can use the learned classifier function $c_\eta$ to obtain a family of gradient estimators for alternative training objectives. For instance, replacing the logit function with the logarithm function yields an objective that minimizes the probability of generated samples being classified as fake. This formulation aligns with GAN (Goodfellow et al., 2014; Nowozin et al., 2016) across different diffusion time steps, which is equivalent to minimizing the *Diffusive Jensen-Shannon (DiJS)* divergence:

$$\nabla_\theta \text{DiJS}(q_\theta(x_0)||p_d(x_0)) \approx \sum_{t=1}^T w(t) \int q_\theta(x_t) \nabla_{x_t} \log(1 - c_\eta(x_t, t)) \frac{\partial x_t}{\partial \theta} dx_t. \quad (18)$$

Alternatively, rather than minimizing the probability of the generated images being fake as used in GAN, one can also maximize the probability of them being real. This approach is referred to as *Diffusive Realism Maximization (DiRM)*, which has the following gradient estimator:

$$\nabla_\theta \text{DiRM}(\theta) \approx -\sum_{t=1}^T w(t) \int q_\theta(x_t) \nabla_{x_t} \log c_\eta(x_t, t) \frac{\partial x_t}{\partial \theta} dx_t. \quad (19)$$

The overall training procedure of our proposed framework is summarized in Algorithm 2. Notably, the DiRM objective—maximizing the likelihood of being real—also mirrors the *non-saturating GAN* formulation (Goodfellow et al., 2014), which is known to provide more stable gradients for the generator compared to the original minimax objective. In principle, once the density ratio is available, any ratio-based divergence measure—such as an $f$-divergence—can be employed to formulate a learning criterion for diffusion distillation. This is reminiscent of the $f$-GAN framework (Nowozin et al., 2016): we will discuss the connection between our proposed method and GAN-based approaches in the next section.

## 4 RELATED WORK

Estimating density ratios is central to many GAN variants (Goodfellow et al., 2014; Nowozin et al., 2016). In classic GANs, the discriminator implicitly estimates the density ratio between real and model distributions. However, for high-dimensional image modeling tasks, both data and model distributions are supported on low-dimensional manifolds and are therefore not absolutely continuous, rendering their densities and the density ratio ill-defined. This further causes the Jensen-Shannon (JS) divergence to be ill-defined and contributes to GAN training instability (Arjovsky & Bottou, 2017; Arjovsky et al., 2017; Mescheder et al., 2018; Roth et al., 2017). To address this, Arjovsky et al. (Arjovsky et al., 2017) proposed to replace the JS divergence with the Wasserstein-1 distance, which can yield meaningful gradients even when the distributions are disjoint. However, training Wasserstein GANs requires enforcing a 1-Lipschitz constraint on the critic, which is challenging in practice and has been approximated using heuristics such as weight clipping (Arjovsky et al., 2017), gradient penalties (Gulrajani et al., 2017), and spectral normalization (Miyato et al., 2018). Despite its theoretical appeal, the divergence minimized in practice often differs from the idealized objective (Mescheder et al., 2018), and that stable GAN training relies more on regularization (e.g., gradient penalties) than on strict divergence minimization (Fedus et al., 2017). Adding Gaussian noise to real and fake samples has been proposed as a way to ensure distributions are fully supported, making the density ratio well-defined (Sønderby et al., 2016; Roth et al., 2017; Nowozin et al., 2016; Zhang et al., 2020). This model-agnostic approach requires no architectural changes but hinges on choosing an effective noise level—something hard to fix throughout training.

Diffusion GAN (Wang et al., 2023) addresses the challenge of selecting a fixed noise level by introducing a diffusion-inspired noise schedule that gradually increases noise in tandem with the model's learning capacity. While this represents the most closely related work to ours, our approach differs in several important aspects. First, diffusion GAN stabilizes training using a StyleGAN-based generator (Karras et al., 2024), which is implicitly trained progressively from low to high resolutions while keeping the network topology fixed. In contrast, our method does not rely on a specialized generator architecture or progressive resolution training. Instead, we adopt a generic U-Net architecture, resulting in a simpler and more broadly applicable framework. Additionally, we do not use common GAN-specific training tricks such as gradient penalties or spectral normalization. Our method provides a clean framework for training one-step diffusion models yet still achieves stable convergence, demonstrating robustness without any additional regularization tricks during training.

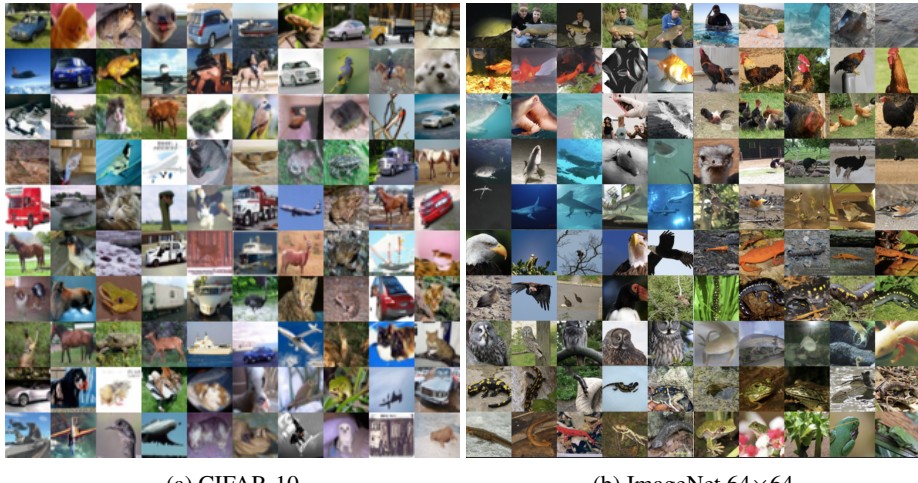

(a) CIFAR-10                           (b) ImageNet $64\times64$

Figure 2: Visualization of sample images generated by DiJS.

Finally, unlike traditional GANs, our framework is no longer adversarial in the strict sense: our generator's performance is *not* dependent on the convergence of a discriminator, which simplifies the training process and mitigates common issues arising from adversarial dynamics (e.g., GANs require a careful balancing between discriminator and generator training).

Because our method does not rely on discriminators, gradient penalties, or adversarial training, we are estimating and minimizing the DiJS divergence—a divergence that remains well-defined even when traditional density ratios remain undefined. This leads to a conceptually cleaner training objective which avoids the complexities and instabilities associated with adversarial min-max optimization. In the next section, we will demonstrate the effectiveness of our method by applying it to train one-step diffusion models for image generation.

## 5 IMAGE GENERATION EXPERIMENTS

We evaluate the performance of our method by training one-step generative models on two standard settings: unconditional generation on the CIFAR-10 ($32\times32$) (Krizhevsky et al., 2009) and class-label-conditioned generation on the ImageNet ($64\times64$) (Deng et al., 2009). Our implementation builds upon the EDM codebase (Karras et al., 2022) and uses the official pre-trained EDM models as teacher models for both datasets. Experiments are conducted on 4 NVIDIA H100 80GB GPUs.

Our one-step student model adopts the same neural network architecture as the teacher model. We use the variance-exploding (VE) noise schedule to define the DiKL divergence for training the student model. Our class-ratio estimator $c_\eta(x_t, t)$ is implemented using the encoder portion of a U-Net to produce a scalar output. This network is approximately half the size of the full U-Net used for score estimation, leading to improved training and memory efficiency. The one-step generator $g_\theta(z)$ is initialized with pre-trained EDM weights from the teacher model with diffusion time $t_{\text{init}} = 2.5$ fixed throughout training and sampling. We follow standard hyperparameter settings for training generative models on CIFAR-10 and ImageNet. Specifically, we set the learning rate to $10^{-5}$, use the weight function $w(t) = \sigma_t^2$, and employ non-leaky data augmentation (Karras et al., 2020) for both datasets. For CIFAR-10, we use a batch size of 64 and an EMA decay rate of 0.5. For ImageNet, all training configurations are set to be the same as those described in Karras et al. (2022).

Interestingly, we observe that performing multiple gradient updates for the class-ratio estimator in each training iteration can accelerate convergence (i.e., reducing the number of training iterations for the student model) without destabilizing the training process, which is distinct from GANs where a careful balancing between the training steps for the discriminator and generator is required for stable training. However, such strategies often increase the overall wall-clock time for training. Hence, we adopt a single-step update strategy for the class-ratio estimator throughout our experiments, consistent with previous works (Luo et al., 2024; Zhou et al., 2024); we leave multi-step class-ratio estimation for future work.

Table 1: Unconditional models on CIFAR10.

| METHOD | NFE ($\downarrow$) | FID ($\downarrow$) | IS ($\uparrow$) |
|---|---|---|---|
| **Teacher model** | | | |
| EDM (Karras et al., 2022) | 35 | 2.04 | 9.84 |
| EDM (Karras et al., 2022) | 1 | 8.70 | 8.49 |
| **Training from scratch (w/o teacher)** | | | |
| CT (Song et al., 2023) | 1 | 8.70 | 8.49 |
| iCT (Song & Dhariwal, 2024) | 1 | 2.83 | 9.54 |
| iCT-deep (Song & Dhariwal, 2024) | 1 | **2.51** | **9.76** |
| BCM (Li & He, 2024) | 1 | 3.10 | 9.45 |
| BCM-deep (Li & He, 2024) | 1 | 2.64 | 9.67 |
| sCT (Lu & Song, 2025) | 1 | 2.85 | - |
| Diffusion-GAN (Wang et al., 2023) | 1 | 3.19 | - |
| IMM (Zhou et al., 2025a) | 1 | 3.20 | - |
| **Distillation w/ teacher init. & supervision** | | | |
| Progressive Distillation (Salimans & Ho, 2022) | 1 | 8.34 | 8.69 |
| DSNO (Zheng et al., 2023) | 1 | 3.78 | - |
| TRACT (Berthelot et al., 2023) | 1 | 3.78 | - |
| CD (Song et al., 2023) | 1 | 3.55 | 9.48 |
| CTM (w/ GAN) (Kim et al., 2024) | 1 | 1.98 | - |
| CTM (w/o GAN) (Kim et al., 2024) | 1 | ¿5.0 | - |
| sCD (Lu & Song, 2025) | 1 | 3.66 | - |
| Diff-Instruct (Luo et al., 2024) | 1 | 4.53 | - |
| ECT (Geng et al., 2025) | 1 | 3.60 | - |
| SiD (Zhou et al., 2024) | 1 | 2.03 | 10.02 |
| SiDA (Zhou et al., 2025b) | 1 | **1.52** | **10.32** |
| **Distillation w/ teacher initialization only** | | | |
| DiRM (ours) | 1 | 4.87 | 9.85 |
| DiKL (ours) | 1 | 3.81 | 9.90 |
| DiJS (ours) | 1 | **2.39** | **9.93** |

Table 2: Class-label-conditioned models on ImageNet 64×64.

| METHOD | NFE ($\downarrow$) | FID ($\downarrow$) |
|---|---|---|
| **Teacher model** | | |
| EDM (Karras et al., 2022) | 79 | 2.44 |
| **Training from scratch (w/o teacher)** | | |
| EDM2-L/XL (Karras et al., 2024) | 1 | 13.0 |
| CT (Song et al., 2023) | 1 | 13.0 |
| iCT (Song & Dhariwal, 2024) | 1 | 4.02 |
| iCT-deep (Song & Dhariwal, 2024) | 1 | 3.25 |
| BCM (Li & He, 2024) | 1 | 4.18 |
| BCM-deep (Li & He, 2024) | 1 | 3.14 |
| sCT (Lu & Song, 2025) | 1 | **2.04** |
| **Distillation w/ teacher init. & supervision** | | |
| Progressive Distillation (Salimans & Ho, 2022) | 1 | 7.88 |
| DSNO (Zheng et al., 2023) | 1 | 7.83 |
| TRACT (Berthelot et al., 2023) | 1 | 7.43 |
| CD (Song et al., 2023) | 1 | 6.20 |
| CTM (w/ GAN) (Kim et al., 2024) | 1 | 1.92 |
| sCD (Lu & Song, 2025) | 1 | 2.44 |
| Diff-Instruct (Luo et al., 2024) | 1 | 5.57 |
| EM Distillation (Xie et al., 2024) | 1 | 2.20 |
| ECT (Geng et al., 2025) | 1 | 2.49 |
| SiD (Zhou et al., 2024) | 1 | 2.02 |
| SiDA (Zhou et al., 2025b) | 1 | **1.35** |
| **Distillation w/ teacher initialization only** | | |
| DiJS (ours) | 1 | **1.54** |

In Tables 1 and 2, we compare our method to previous methods for training one-step generative models. To highlight methodological differences, we categorize these approaches into three groups: (1) training from scratch (e.g., Diffusion-GAN, trajectory-based distillation), (2) training with teacher initialization and supervision (e.g., using teacher score, denoiser or ODE solvers in the loss), and (3) training with teacher weight initialization only (our method). We find that our proposed method, DiJS, achieves competitive one-step generation performance despite not using any teacher score information, outperforming most state-of-the-art distillation methods that rely on full teacher supervision. The only method that outperforms ours is SiDA (Zhou et al., 2025b), which depends on training data, teacher score supervision, teacher weight initialization, and student score estimation. In contrast, our method requires only training data, teacher weight initialization, and class-ratio estimation. Notably, this eliminates the need for teacher supervision in the training process. Also, class-ratio estimation is both simpler and more lightweight than student score estimation, as the ratio network is approximately half the size of a full score network. This results in a more streamlined and memory-efficient training framework.

# 6 ANALYSIS OF THE ROLE OF TEACHER WEIGHT INITIALIZATION

In the previous section, our one-step student model was initialized with the teacher model's weights. We observed that training from random initialization led to mode collapse; see Figure 3c for an example. One possible explanation is that mode collapse arises from the training objectives (i.e., reverse KL or JS divergence), a phenomenon also observed in GAN literature (Goodfellow et al., 2014). To understand why initializing the student model with the teacher model's weights prevents mode collapse in the training process, we investigate the following two hypotheses.

**Function Space Hypothesis.** *Teacher weight initialization provides a more structured latent-to-output functional mapping, i.e., different locations in the latent space are initially mapped to distinct images, preventing mode collapse.*

This hypothesis originally arose from visualizing initialized samples as shown in Figure 3a, showing that initialization already induces diverse mappings, with the student model training stage primarily refining these initializations into sharper images. Somewhat surprisingly, however, we find that functional initialization alone is insufficient to prevent mode collapse. To show this, instead of training the teacher model across different diffusion time steps $t$ and selecting a single time step $t_{\text{init}}$

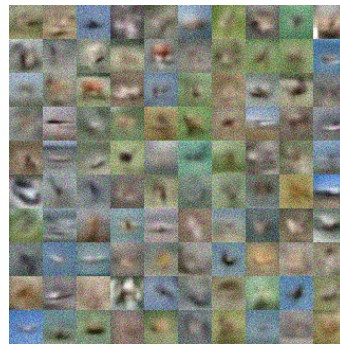 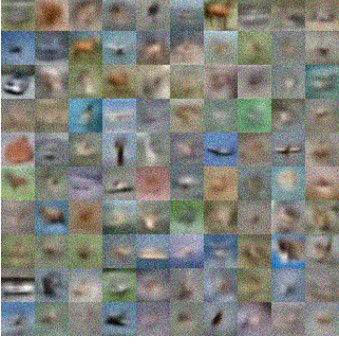 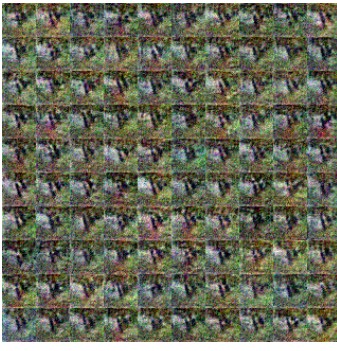

(a) Single-level DSM Init.          (b) Multi-level DSM Init.          (c) Collapsed Samples

Figure 3: Visualization of different initializations and collapsed samples on CIFAR-10.

for initialization, we only pre-train the teacher model at the selected time step $t_{\text{init}}$ and use its weight to initialize the one-step student model. This setup ensures identical latent-to-output mappings for the student model at initialization as shown in Figure 3b. However, with this initialization, the student model still exhibits mode collapse early in the student model training stage, which suggests that the functional mapping perspective alone does not fully explain the mode-collapse issue.

**Feature Space Hypothesis.** *Teacher weight initialization provides a rich set of multi-level features learned when pre-training the teacher diffusion model, which help prevent mode collapse.*

To verify this hypothesis and isolate the effects of learned functional features, we pre-train the teacher model on CIFAR-100 while excluding all classes that overlap with CIFAR-10. This ensures that images from the target classes that the student model aims to generate are absent during pre-training, allowing us to focus solely on the contribution of the learned features. We train the teacher model using increasingly larger subsets of CIFAR-100 with (10, 50, 90)

Table 3: Performance of one-step models trained by DiJS with different initializations on various CIFAR subsets.

| Initialization method | Initialization dataset | FID |
| --- | --- | --- |
| No initialization | - | collapsed |
| Single-level DSM | full CIFAR-10 | collapsed |
| Multi-level DSM | 10 classes in CIFAR-100 | collapsed |
| | 50 classes in CIFAR-100 | 6.20 |
| | 90 classes in CIFAR-100 | 6.01 |
| | full CIFAR-10 | 2.39 |

classes, creating a setting with increasing feature diversity. Table 3 shows the performance of our one-step student model on CIFAR-10 initialized with the weights of teacher models trained on varying numbers of CIFAR-100 classes. We find that when the teacher model is trained on only 10 classes, mode collapse still occurs. However, as the number of training classes increases, the student model no longer collapses, indicating that feature richness plays a crucial role in preventing mode collapse. Nevertheless, despite mitigating mode collapse, this initialization strategy achieves an FID of 6.01 when the teacher model is pre-trained on all 90 non-overlapping classes in CIFAR-100, which is significantly worse than the FID (2.39) obtained when directly using CIFAR-10 as the pre-training dataset. This suggests that while feature richness is essential for stabilizing training, functional mapping initialization remains important for achieving higher sample quality.

## 7 CONCLUSIONS

We studied whether one-step diffusion models can be trained without a pre-trained teacher. To this end, we introduced score-free training methods based on class-ratio estimation, eliminating the need for teacher or student score supervision. Our method matched the quality of state-of-the-art teacher-supervised approaches while reducing complexity and memory usage. It also simplified GAN-style training by removing adversarial tricks, showing that a single time-conditioned class-ratio estimator suffices for stable training. A key finding was that while teacher score supervision is unnecessary, teacher-based weight initialization remains important—not for better mappings, but for the multi-level features learned across noise levels, which help prevent mode collapse. Future directions include unsupervised or self-supervised pretraining for rich initializations and extending the framework to modalities like audio and video.

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

## A VALIDITY OF DIFFUSIVE DIVERGENCE

We follow the original Spread Divergence (Zhang et al., 2020) and provide a simple proof of the validity of the diffusive KL (DiKL) divergence. The extension to the diffusive Jensen-Shannon (DiJS) divergence is straightforward. See Zhang et al. (2020) for a generalized proof that includes cases with non-absolutely continuous distributions and non-Gaussian kernels.

Our goal is to show that for the DiKL defined as

$$\mathrm{DiKL}_{\mathcal{K}}(q_\theta(x_0) \,\|\, p_d(x_0)) = \sum_{t=1}^{T} w(t)\,\mathrm{KL}(q_\theta(x_t) \,\|\, p_d(x_t)), \tag{20}$$

where $w(t) > 0$, and the densities $q_\theta(x_t)$ and $p_d(x_t)$ represent the noisy model and data distributions respectively, defined as:

$$q_\theta(x_t) = \int q_\theta(x_0)\,k_t(x_t \mid x_0)\,dx_0, \tag{21}$$

$$p_d(x_t) = \int p_d(x_0)\,k_t(x_t \mid x_0)\,dx_0, \tag{22}$$

with $k_t(x_t \mid x_0)$ denoting the transition kernel (e.g., Gaussian noise). Then:

$$\mathrm{DiKL}_{\mathcal{K}}(q_\theta(x_0) \,\|\, p_d(x_0)) = 0 \iff q_\theta(x_0) = p_d(x_0).$$

Since $w(t) > 0$ and the KL divergence is non-negative, it suffices to show that:

$$\mathrm{KL}(q_\theta(x_t) \,\|\, p_d(x_t)) = 0 \iff q_\theta(x_t) = p_d(x_t) \iff q_\theta(x_0) = p_d(x_0).$$

To demonstrate this, assume that $k_t(\epsilon) = \mathcal{N}(0, \sigma^2 I)$, and rewrite the noisy densities as convolutions:

$$q_\theta(x_t) = (q_\theta * k_t)(x_t), \tag{23}$$
$$p_d(x_t) = (p_d * k_t)(x_t). \tag{24}$$

Suppose $q_\theta(x_t) = p_d(x_t)$. Applying the Fourier transform $\mathcal{F}$, we obtain:

$$\mathcal{F}(q_\theta * k_t) = \mathcal{F}(q_\theta) \cdot \mathcal{F}(k_t), \tag{25}$$
$$\mathcal{F}(p_d * k_t) = \mathcal{F}(p_d) \cdot \mathcal{F}(k_t). \tag{26}$$

Given $q_\theta(x_t) = p_d(x_t)$, we have:

$$q_\theta(x_t) = p_d(x_t) \iff \mathcal{F}(q_\theta) \cdot \cancel{\mathcal{F}(k_t)} = \mathcal{F}(p_d) \cdot \cancel{\mathcal{F}(k_t)} \iff \mathcal{F}(q_\theta) = \mathcal{F}(p_d) \iff q_\theta = p_d.$$

Therefore, $\mathrm{KL}(q_\theta(x_t) \,\|\, p_d(x_t)) = 0 \iff q_\theta(x_t) = p_d(x_t)$, and thus:

$$\mathrm{DiKL}_{\mathcal{K}}(q_\theta(x_0) \,\|\, p_d(x_0)) = 0 \iff q_\theta(x_0) = p_d(x_0).$$

## B DERIVATION OF ANALYTICAL GRADIENT FOR REVERSE KL

The gradient of reverse DiKL w.r.t. the model parameter $\theta$ is given by

$$\nabla_\theta \mathrm{DiKL}_{k_t}(q_\theta \| p_d) = \int q_\theta(x_t)\,(\nabla_{x_t} \log q_\theta(x_t) - \nabla_{x_t} \log p_d(x_t))\,\frac{\partial x_t}{\partial \theta}\,dx_t. \tag{27}$$

The reverse DiKL at time $t$ is defined as

$$\mathrm{DiKL}_{k_t}(q_\theta \| p_d) = \int (\log q_\theta(x_t) - \log p_d(x_t))\,q_\theta(x_t)\,dx_t. \tag{28}$$

We first reparameterize $x_t$ as a function of $z$ and $\epsilon$:

$$x_t = \alpha_t g_\theta(z) + \sigma_t \epsilon_t \equiv h_\theta(z, \epsilon_t), \tag{29}$$

where $z \sim p(z) \equiv \mathcal{N}(z|0, I)$ and $\epsilon_t \sim p(\epsilon_t) \equiv \mathcal{N}(\epsilon_t|0, I)$. It then follows that

$$
\begin{aligned}
\nabla_\theta \mathrm{DiKL}_{k_t}(q_\theta \| p_d) &= \nabla_\theta \int \left( \log q_\theta(x_t) - \log p_d(x_t) \right) q_\theta(x_t) dx_t \\
&= \nabla_\theta \iiint \left( \log q_\theta(x_t) - \log p_d(x_t) \right) \delta(x_t - h_\theta(z, \epsilon_t)) p(z) p(\epsilon_t) dx_t dz d\epsilon \\
&= \nabla_\theta \iint \left( \log q_\theta(x_t) - \log p_d(x_t) \right) |_{x_t = h_\theta(z, \epsilon_t)} p(z) p(\epsilon_t) dz d\epsilon \\
&= \int \left( \nabla_\theta \log q_\theta(x_t) + \nabla_{x_t} \log q_\theta(x_t) \frac{\partial x_t}{\partial \theta} - \nabla_{x_t} \log p_d(x_t) \frac{\partial x_t}{\partial \theta} \right) p_\theta(x_t) dx_t \\
&= \int \left( \nabla_{x_t} \log q_\theta(x_t) \frac{\partial x_t}{\partial \theta} - \nabla_{x_t} \log p_d(x_t) \frac{\partial x_t}{\partial \theta} \right) p_\theta(x_t) dx_t,
\end{aligned}
$$

where the last line follows since

$$
\int \nabla_\theta \log q_\theta(x_t) q_\theta(x_t) dx_t = \int \nabla_\theta q_\theta(x_t) dx_t = \nabla_\theta \int q_\theta(x_t) dx_t = \nabla_\theta 1 = 0. \tag{30}
$$

This completes the proof.

## C CLASS RAITIO ESTIMATION FOR DiJS DIVERGENCE

We define the diffusive Jensen-Shannon (DiJS) divergence between the model distribution $q_\theta(x_0)$ and the data distribution $p_d(x_0)$ as:

$$
\mathrm{DiJS}_{\mathcal{K}}(q_\theta(x_0) \| p_d(x_0)) = \sum_{t=1}^{T} w(t) \, \mathrm{JS}(q_\theta(x_t) \| p_d(x_t)), \tag{31}
$$

where $w(t) > 0$ are positive weights, and the noisy distributions $q_\theta(x_t)$ and $p_d(x_t)$ are defined via convolution with a transition kernel $k_t(x_t \mid x_0)$ (e.g., Gaussian noise):

$$
q_\theta(x_t) = \int q_\theta(x_0) \, k_t(x_t \mid x_0) \, dx_0, \tag{32}
$$

$$
p_d(x_t) = \int p_d(x_0) \, k_t(x_t \mid x_0) \, dx_0. \tag{33}
$$

The Jensen-Shannon divergence between two distributions $q$ and $p$ is given by:

$$
\mathrm{DiJS}(q_\theta \| p_d) = \frac{1}{2} \mathrm{DiKL}(q_\theta \| \frac{1}{2}(q_\theta + p_d)) + \frac{1}{2} \mathrm{DiKL}(p_d \| \frac{1}{2}(q_\theta + p_d)). \tag{34}
$$

We now derive the gradient of the DiJS divergence with respect to model parameters $\theta$. By the chain rule:

$$
\nabla_\theta \mathrm{DiJS}(q_\theta(x_0) \| p_d(x_0)) = \sum_{t=1}^{T} w(t) \, \nabla_\theta \mathrm{JS}(q_\theta(x_t) \| p_d(x_t)). \tag{35}
$$

Assume we obtain the optimal classifier $c^*(x_t, t) \equiv p(y = 1 \mid x_t, t)$. We follow the GAN method (Goodfellow et al., 2014) to ignore the second term when the class ratio estimation is optimal, the first term in the JS divergence (involving $q_\theta$) gives:

$$
\nabla_\theta \mathrm{JS}(q_\theta(x_t) \| p_d(x_t)) \approx \int \nabla_\theta q_\theta(x_t) \log \left( \frac{q_\theta(x_t)}{m(x_t)} \right) dx_t \tag{36}
$$

$$
= \int q_\theta(x_t) \nabla_{x_t} \log \left( \frac{q_\theta(x_t)}{m(x_t)} \right) \frac{\partial x_t}{\partial \theta} dx_t, \tag{37}
$$

where $m(x_t) = \frac{1}{2}(q_\theta(x_t) + p_d(x_t))$. Using the class-ratio view, we substitute:

$$
\frac{q_\theta(x_t)}{m(x_t)} = \frac{m(x_t|y = 0)}{m(x_t|y = 0) + m(x_t|y = 1)} = 1 - c^*(x_t, t),
$$

which gives:

$$\nabla_\theta \text{DiJS}(q_\theta(x_0) \| p_d(x_0)) \approx \sum_{t=1}^{T} w(t) \int q_\theta(x_t) \nabla_{x_t} \log(1 - c^*(x_t, t)) \frac{\partial x_t}{\partial \theta} dx_t. \qquad (38)$$

If we don't follow the GAN (Goodfellow et al., 2014) approximation, one can also treat this gradient estimation as an exact estimation of a DiKL with a mixture target distribution

$$\nabla_\theta \text{DiKL}\left(q_\theta \| \frac{1}{2}(q_\theta + p_d)\right) = \sum_{t=1}^{T} w(t) \int q_\theta(x_t) \nabla_{x_t} \log(1 - c^*(x_t, t)) \frac{\partial x_t}{\partial \theta} dx_t.$$

This is also a valid divergence between $q_\theta$ and $p_d$ since $q_\theta = \frac{1}{2}q_\theta + \frac{1}{2}p_d \iff q_\theta = p_d$.

# D    ADDITIONAL IMAGE GENERATION RESULTS

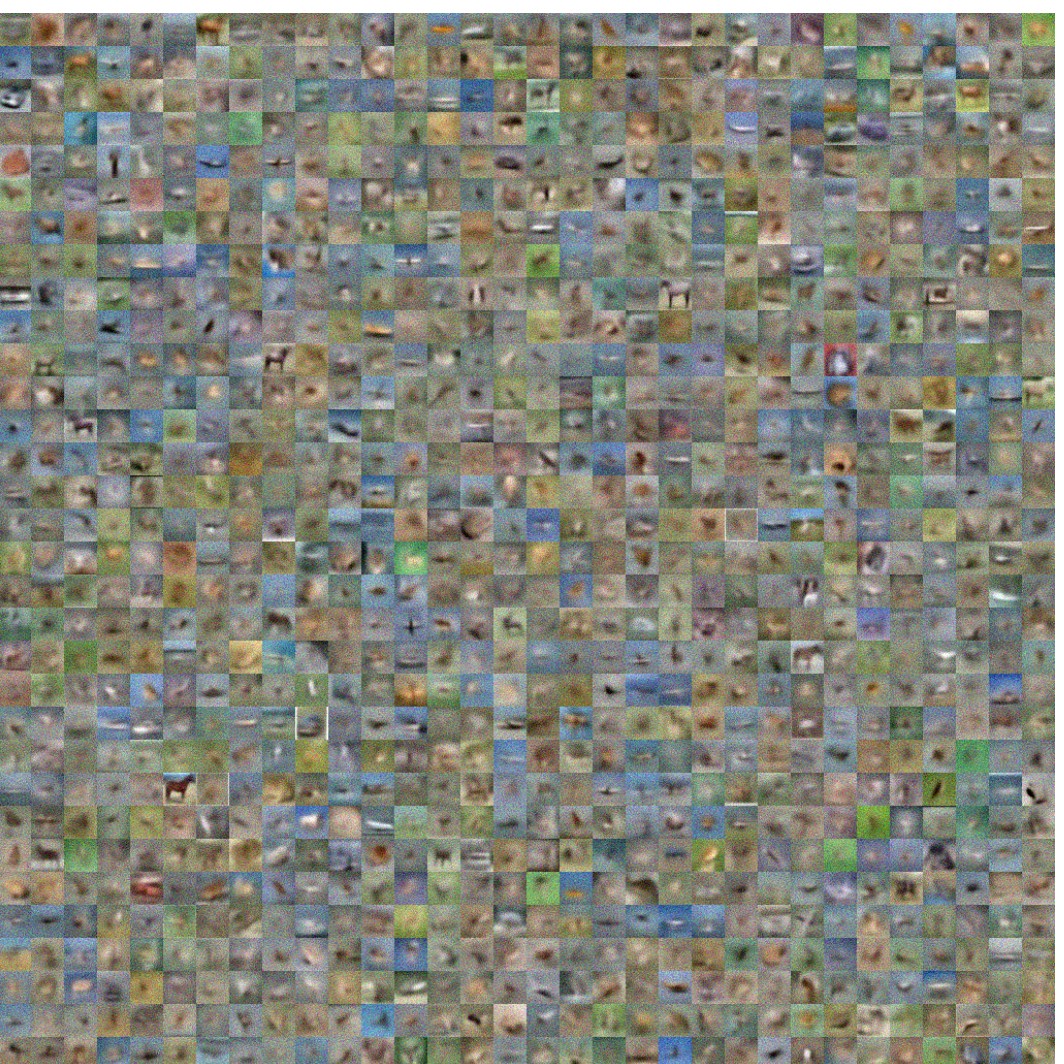

Figure 4: Visualization of the samples from the multi-level DSM Initialization

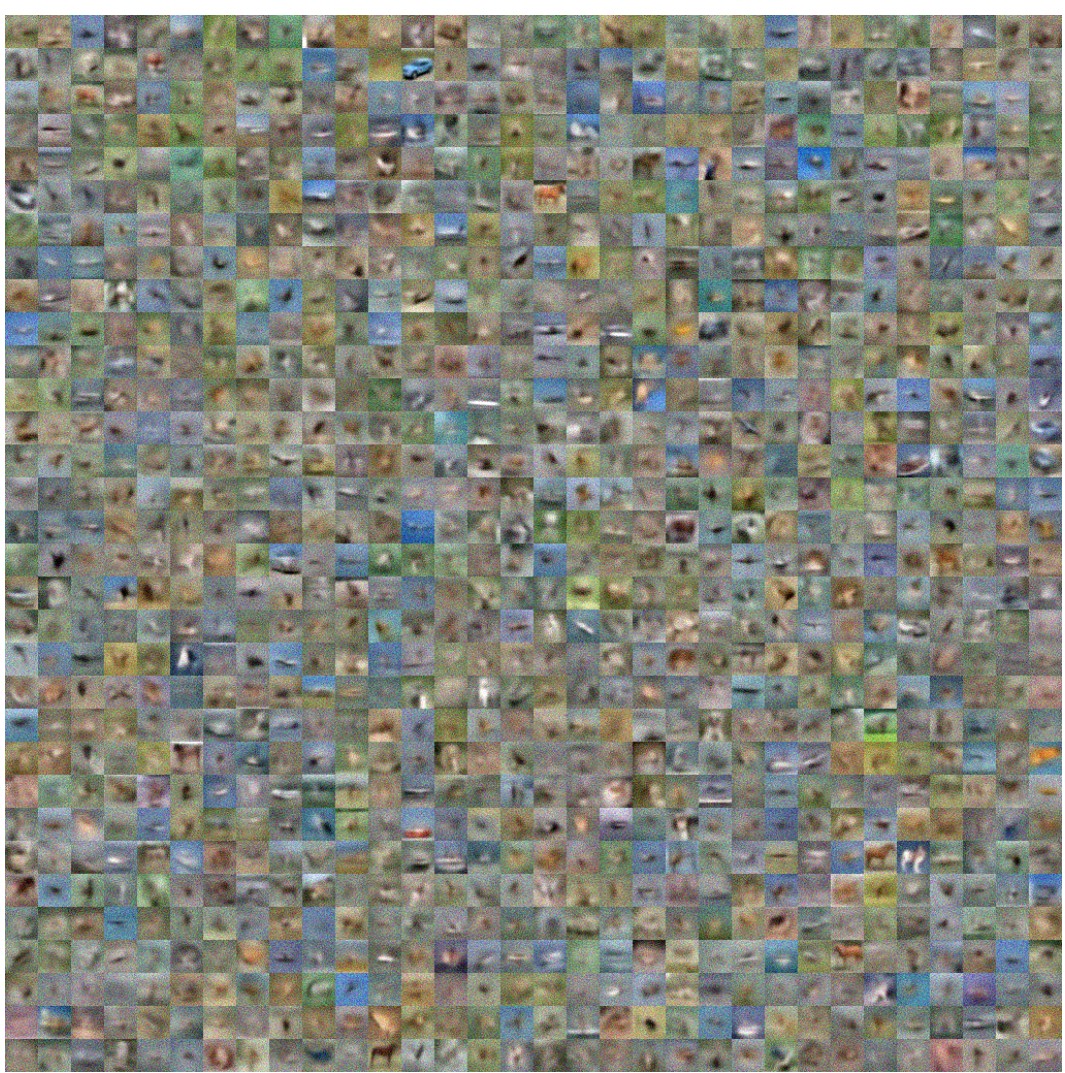

Figure 5: Visualization of the samples from the single-level DSM Initialization

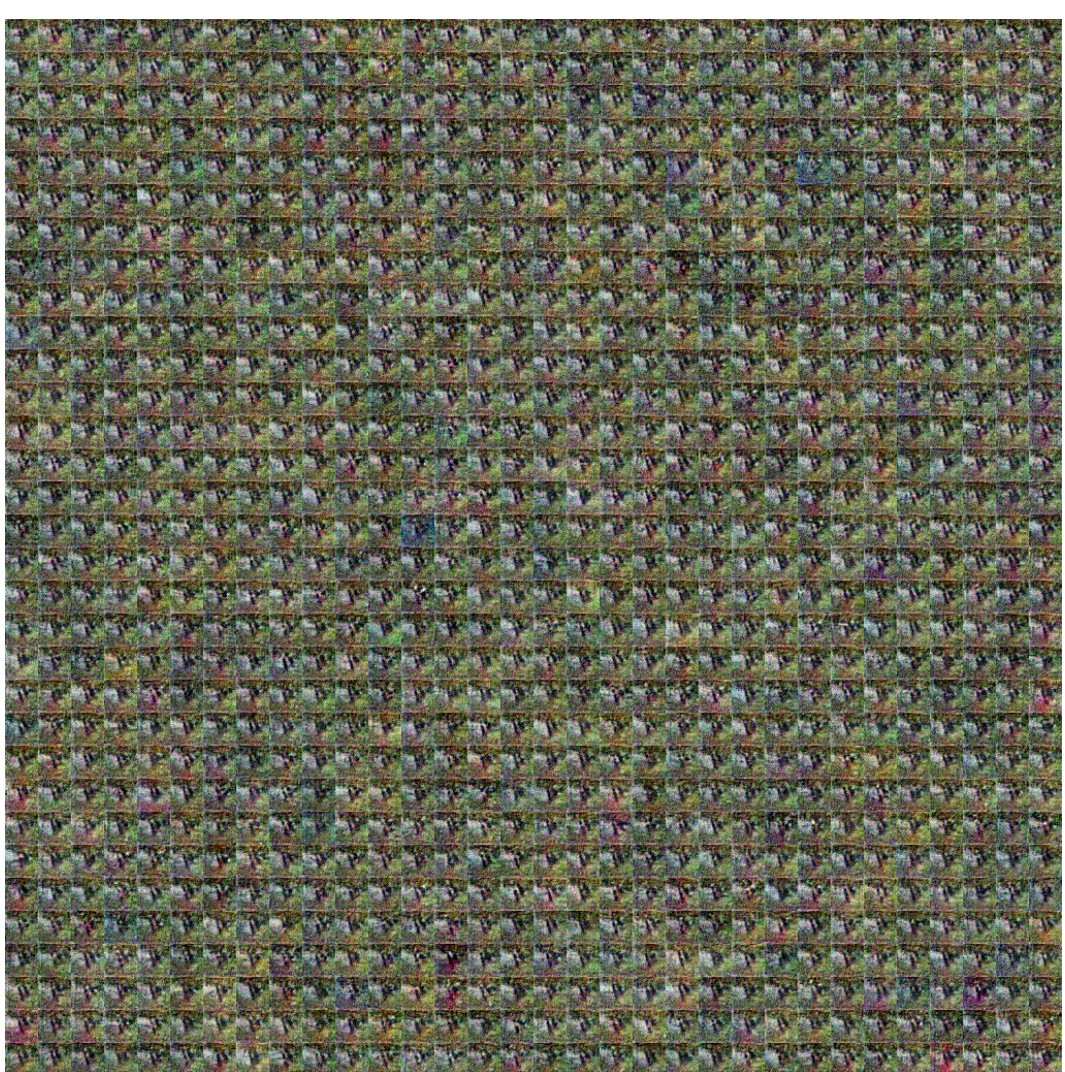

Figure 6: Visualization of the collapsed samples

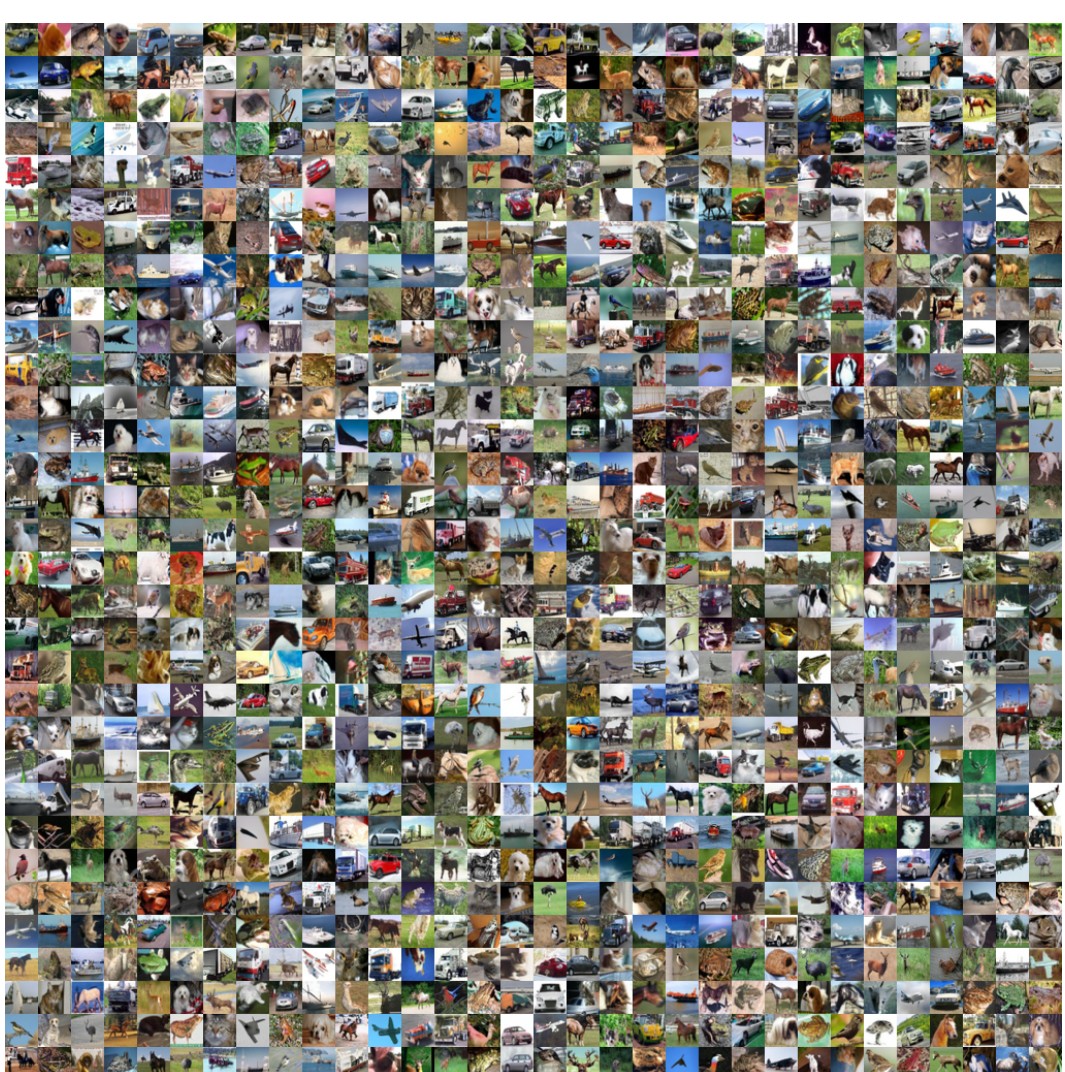

Figure 7: CIFAR Visualization of the DiJS samples (FID=2.39, IS=9.93)

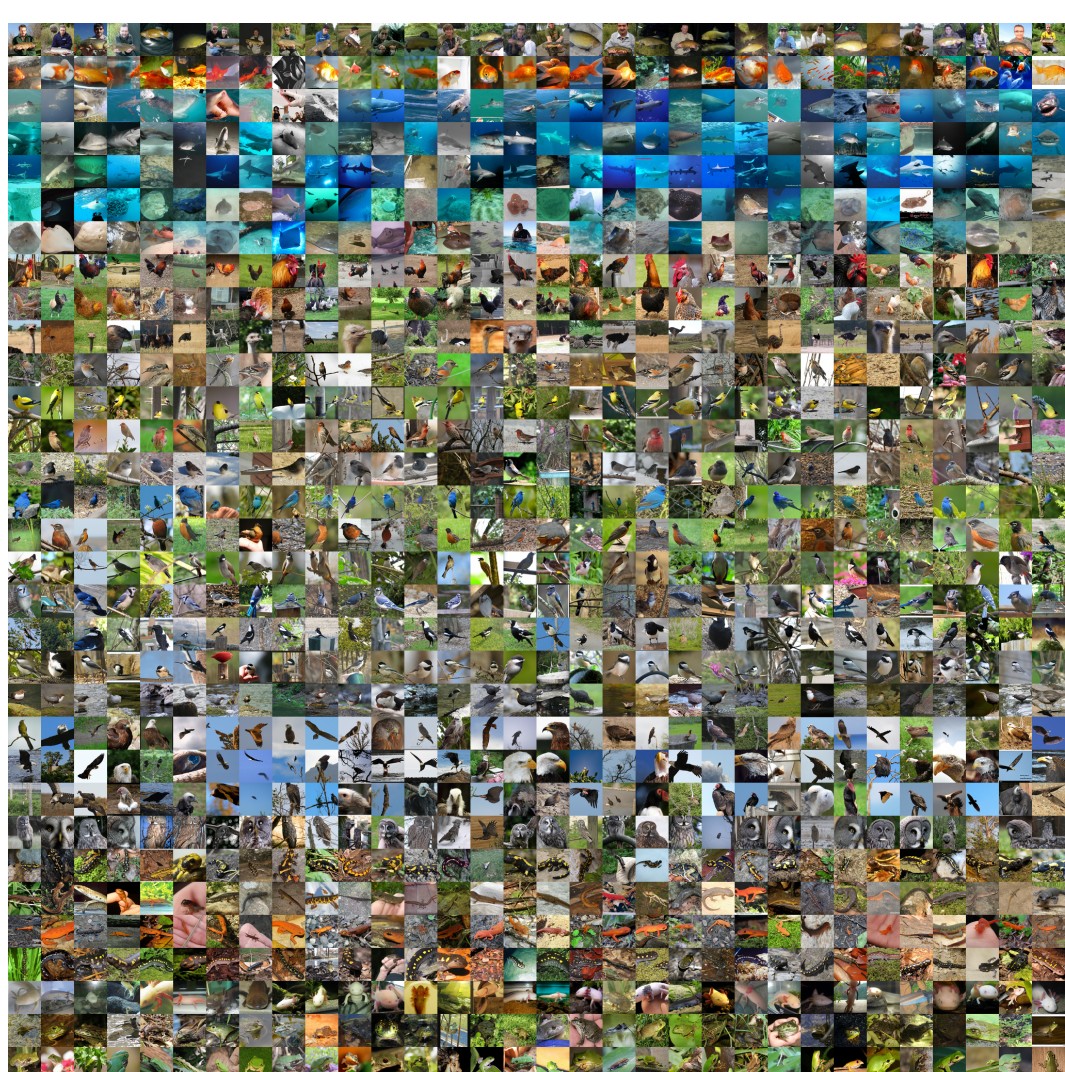

Figure 8: ImageNet 64x64 Visualization of the DiJS samples (FID=1.54)

