# OpenReview forum: "Towards Training One-Step Diffusion Models Without Distillation"
_ICLR.cc/2026/Conference — ICLR 2026 Conference Withdrawn Submission_

### Official Review · Reviewer_weJd · 2025-10-30

**Soundness:** 2
**Presentation:** 3
**Contribution:** 3
**Rating:** 4
**Confidence:** 4

**Summary:**

This paper investigates the possibility of training one-step diffusion models without teacher supervision, i.e., training on the score matching gradient. Instead, it proposes a GAN-like method that trains a classifier to distinguish between data from noisy real data or noisy synthetic data. Performances on CIFAR-10 and ImageNet are comparable with those trained under teacher supervision. In summary, this paper proposes a novel one-step diffusion method.

**Strengths:**

The proposed method is a simple yet effective method. It gets rid of teacher supervision but can achieve comparable performances. From the efficiency perspective, the contribution of this method is significant.

**Weaknesses:**

The paper is missing some details and lacks experimental analysis. The major weaknesses are listed below. Please also see the questions.

1. How is the class-ratio estimator trained? I didn't find details of this paper, and I hope the author can clarify.
2. Based on the performances in Tables 1 and 2, I would say the major advantage of this method is its efficiency, since it doesn't need the teacher's output. Some comparison of the efficiency can make this paper more convincing.
3. The author only performs experiments on class-conditional image generation. How about the text-to-image task?

Minors: Figure 1 has overlapping legends, and there's a typo in Table 1 on Line 393.

I'll consider raising the score if all conerns are addressed.

**Questions:**

1. Is the class-ratio estimator pretrained, or is it trained jointly with the diffusion model? I'd like to see some analysis on the design choice of training the class-ratio estimator.
2. Can your method scale up to more complex tasks, such as video generation?

---

### Official Review · Reviewer_vXEj · 2025-10-31

**Soundness:** 1
**Presentation:** 2
**Contribution:** 1
**Rating:** 2
**Confidence:** 5

**Summary:**

This manuscript proposes to train an one-step inference model not trained by classical diffusion process and distillation. It highlights the initialization of well-trained diffusion models, and claims that it exactly matter that features cover varied levels of noise.

**Strengths:**

- A thorough background introduction spanning at least four pages is provided without appendix statements, and only one page of methodology in the main paper and two pages of experiments to support main claims with tremendous and pure visualization in the appendix, catering to readers unfamiliar with these foundational concepts.

**Weaknesses:**

- Limited experiments on only small-scale datasets. The experimental parts of initialization just propose two hypotheses, but limited additional experiments, generalization ablation and analyses are provided to support them.
- It would be better to compare the proposed methods on similar settings with GANs, where the exact framework is very similar to GANs', that of one-step inference and without classic diffusion model designs of distillation, but a score-parameterization GAN.
- The incomplete post-ablation model serves as the baseline, e.g., EDM-2-XL of 13.0 FID. And the one of the main claims says that one-step inference beats most of the other diffusion models under these settings.

**Questions:**

- Even if the author claims that this is not a GAN that need converged discriminator in page 7, it's still hard to convince. Isn't it a better statement that it's a score-parameterization GAN initialized by diffusion model's weights under Bayesian formulation, where the so-called classifier is the discriminator to learn from the teacher models?

---

### Official Review · Reviewer_Anmi · 2025-11-01

**Soundness:** 2
**Presentation:** 2
**Contribution:** 2
**Rating:** 4
**Confidence:** 4

**Summary:**

This paper proposes a novel diffusion model training method that directly trains a one-step student model without supervision from a teacher model.

**Strengths:**

The authors adopt a likelihood ratio estimation approach to directly estimate the density ratio between the student and teacher models, thereby avoiding explicit supervision from the teacher model. This is a novel training strategy.

**Weaknesses:**

1. The authors claim in the introduction that existing methods cannot achieve high-quality generation within five steps. However, to my knowledge, methods such as DMD2, SenseFlow, Hybrid SD, and Consistency Trajectory Model have already demonstrated high-quality generation within four steps or even a single step.

2. Although the proposed method does not require training two models simultaneously like VSD, it still requires a pre-trained classifier before training the student model. Therefore, in total, the proposed approach still involves training two models—a classifier and a student model—which does not seem to offer a clear efficiency advantage over the original VSD.

3. Typically, the related work section is placed either after the introduction or after the experiments but positioning it between the method and experiments sections seems inappropriate.

4. The experiments are only conducted on CIFAR-10 and ImageNet, which are relatively simple image datasets. More complex tasks, such as text-to-image generation, are not explored. Given that text-to-image synthesis is currently the most widely applied domain for diffusion models, it is necessary to include experiments on this task and compare with state-of-the-art models such as SD 3.5, SDXL, and FLUX.

**Questions:**

Please see weakness.

---

### Note · Authors · 2025-11-12

**Comment:**

Thank you for the useful feedback. We have decided to withdraw the paper for further improvement.

**Withdrawal Confirmation:**

I have read and agree with the venue's withdrawal policy on behalf of myself and my co-authors.